# Injectable Photocrosslinked Hydrogel Dressing Encapsulating Quercetin-Loaded Zeolitic Imidazolate Framework-8 for Skin Wound Healing

**DOI:** 10.3390/pharmaceutics16111429

**Published:** 2024-11-10

**Authors:** Zhao Chen, Man Zhe, Wenting Wu, Peiyun Yu, Yuzhen Xiao, Hao Liu, Ming Liu, Zhou Xiang, Fei Xing

**Affiliations:** 1Department of Orthopedic Surgery, Orthopedic Research Institute, West China Hospital, Sichuan University, Chengdu 610041, China; 15620610661@163.com (Z.C.); emailoflh@163.com (H.L.); liuminglm15@163.com (M.L.); 2Animal Experiment Center, West China Hospital, Sichuan University, Chengdu 610041, China; zheman@wchscu.cn; 3Department of Pediatric Surgery, Division of Orthopedic Surgery, Orthopedic Research Institute, Laboratory of Stem Cell and Tissue Engineering, State Key Laboratory of Biotherapy, West China School of Medicine, West China Hospital, Sichuan University, Chengdu 610041, China; 2022141460338@stu.edu.cn; 4LIMES Institute, Department of Molecular Brain Physiology and Behavior, University of Bonn, Carl-Troll-Str. 31, 53115 Bonn, Germany; s49pyu@uni-bonn.de; 5Institute of Basic Medical Sciences Chinese Academy of Medical Sciences, School of Basic Medicine Peking Union Medical College, Beijing 100005, China; s2024005012@student.pumc.edu.cn

**Keywords:** wound healing, hydrogel wound dressings, antibacterial, immunomodulation, angiogenesis

## Abstract

**Background**: Wound management is a critical component of clinical practice. Promoting timely healing of wounds is essential for patient recovery. Traditional treatments have limited efficacy due to prolonged healing times, excessive inflammatory responses, and susceptibility to infection. **Methods**: In this research, we created an injectable hydrogel wound dressing formulated from gelatin methacryloyl (GelMA) that encapsulates quercetin-loaded zeolitic imidazolate framework-8 (Qu@ZIF-8) nanoparticles. Next, its ability to promote skin wound healing was validated through in vitro experiments and animal studies. **Results**: Research conducted both in vitro and in vivo indicated that this hydrogel dressing effectively mitigates inflammation, inhibits bacterial growth, and promotes angiogenesis and collagen synthesis, thus facilitating a safe and efficient healing process for wounds. **Conclusions**: This cutting-edge scaffold system provides a novel strategy for wound repair and demonstrates significant potential for clinical applications.

## 1. Introduction

The skin, being the largest organ in the human body, is vital for maintaining internal homeostasis and preventing the invasion of harmful substances and pathogenic microorganisms [1,2]. When the skin is damaged, the healing of wounds is essential for restoring the integrity of the epidermal barrier [3]. Generally, wound healing is a multifaceted process influenced by various factors. Persistent inflammation, vascular dysfunction, and bacterial infection are potentially major contributors to delayed healing [4,5]. Currently, treatment for skin wounds includes surgical debridement, grafting, dressing changes, hyperbaric oxygen therapy, and more [4]. Among these, dressing changes have become the primary strategy for wound treatment due to their low cost and ease of application. However, in clinical practice, traditional dressings often struggle to sustain the moist conditions essential for optimal wound healing [6]. Additionally, changing dressings can cause secondary trauma to newly formed granulation tissue, adversely affecting healing efficiency and outcomes while significantly increasing the workload of healthcare professionals [7]. Thus, there is a pressing need for a wound treatment strategy that balances simplicity and high effectiveness.

Metal–organic frameworks (MOFs) are crystalline polymers created through the coordination of metal ions with organic ligands [8]. These materials exhibit exceptionally high specific surface areas, adjustable nanostructures, good biocompatibility, and evenly distributed catalytic sites, rendering them particularly advantageous for biomedical applications [9,10]. Among various MOFs, zeolitic imidazolate framework-8 (ZIF-8) has gained considerable interest from researchers in the field of tissue engineering due to its excellent biocompatibility [11,12]. Studies have demonstrated that ZIF-8 not only exhibits multiple biological effects, including immunomodulation [13], antibacterial properties [14], and promotion of angiogenesis [15], but it also possesses outstanding drug-loading capacity [16], making it an ideal drug carrier for precise targeted delivery, thereby greatly expanding its application range in drug therapy.

Quercetin, a naturally occurring flavonoid widely found in plants, has garnered considerable interest from researchers due to its remarkable anti-inflammatory and antibacterial properties [17,18]. However, its inherent low water solubility and rapid metabolism in vivo limit its widespread application in the biomedical field. To overcome these limitations, researchers have attempted to load quercetin onto drug carriers such as nanoparticles for its sustained release. For example, Bakr et al. loaded quercetin onto chitosan nanoparticles, which effectively improved the bioavailability of quercetin. Additionally, it alleviated cisplatin-induced renal and testicular toxicity by inhibiting oxidative stress, inflammation, and apoptosis [19]. Inspired by the above research, this study aims to encapsulate quercetin within ZIF-8 nanoparticles to prepare Qu@ZIF-8 nanoparticles, leveraging the unique advantages of ZIF-8 to enhance drug stability and control drug release, with the goal of improving the bioavailability of quercetin and effectively prolonging its action time in vivo. Nevertheless, the Qu@ZIF-8 nanoparticles alone may not provide an optimal repair environment for wound healing. Therefore, we propose the construction of a biocompatible carrier capable of loading Qu@ZIF-8 nanoparticles, which will not only create a suitable environment for wound healing but also facilitate the sustained release of Qu@ZIF-8.

Hydrogels are three-dimensional porous structures made from hydrophilic polymers, with interconnected pore structures that facilitate nutrient transport essential for cell growth and the removal of cellular metabolic waste, as well as promote intercellular communication [20]. Notably, hydrogels possess excellent hydrophilicity, allowing them to absorb exudates from wounds and preserve a humid environment around the wound region [21]. Due to their injectable nature, hydrogels can conform closely to various wound morphologies, offering a highly adaptable protective barrier. Furthermore, hydrogels demonstrate significant potential as drug carriers, allowing for precise and localized delivery of therapeutics to targeted tissue areas, which enhances treatment efficacy and reduces systemic side effects [22]. To replicate the properties of the native extracellular matrix and establish a suitable environment for cell growth and tissue regeneration, researchers synthesized a photoactive gelatin derivative, gelatin methacryloyl (GelMA), from the hydrolysis products of collagen. The GelMA can undergo photopolymerization in the presence of a photoinitiator to form hydrogels, ensuring its convenience.

Moreover, the arginine–glycine–aspartic acid (RGD) sequences in GelMA ensure excellent biocompatibility [23], while the matrix metalloproteinase (MMP) sites in its structure confer biodegradability [24]. Given these advantages, many studies have utilized GelMA hydrogels to construct wound dressings, achieving satisfactory repair results. For instance, in the research conducted by Nascimento et al., GelMA hydrogels containing Punica granatum extract were able to promote the differentiation of myofibroblasts, thereby facilitating wound closure [25]. Moreover, in the study by Yi et al., GelMA hydrogels were directly encapsulated with quercetin, which successfully reduced oxidative stress and promoted skin wound healing [26]. Therefore, in this study, we planned to load Qu@ZIF-8 nanoparticles into GelMA hydrogels to construct a novel skin wound dressing.

Although various hydrogel-based wound dressing materials have been developed to promote wound healing; however, from a clinical perspective, there are still relatively few wound dressings that can simultaneously ensure ease of application while possessing the characteristics of immunomodulation, infection prevention, and angiogenesis promotion. In this study, we achieved the dual sustained release of quercetin and zinc ions by embedding quercetin-loaded ZIF-8 nanoparticles within GelMA hydrogels. This innovative design effectively mitigates the toxic side effects associated with drug burst release and significantly prolongs the effective duration of the drug’s action. Experimental results indicate that the hydrogel dressing exhibits remarkable antibacterial, immunomodulatory, and angiogenic activities. Additionally, the photopolymerization properties of the hydrogel allow it to maintain injectability while providing sufficient mechanical strength, highlighting its substantial application value in treating skin wounds (Figure 1).

## 2. Materials and Methods

Materials: Poly(ethylene glycol) diacrylate (PEGDA), gelatin, methacrylic anhydride, quercetin, 2-methylimidazole, Zn(NO_3_)·6H_2_O, and lithium phenyl-2,4,6-trimethylbenzoylphosphinate (LAP) were procured from Sigma-Aldrich (St. Louis, MO, USA). The cell counting kit-8 (CCK-8), Calcein-AM/PI staining kit, Actin-tracker staining kit, and DAPI staining solution were sourced from Beyotime (Shanghai, China). Serum and culture media were provided by Gibco (Grand Island, NY, USA). The cell lines were all sourced from the Sunn Biotechnology Co., Ltd. (Hangzhou, China). The matrigel matrix was acquired from Corning Inc. (Corning, NY, USA). The reverse transcription and cDNA amplification reagents were sourced from Vazyme (Nanjing, China). Antibodies for ELISA, immunofluorescence, and immunohistochemistry were obtained from Abcam (Cambridge, UK).

Preparation of ZIF-8 and Qu@ZIF-8 Nanoparticles: ZIF-8 and quercetin-loaded ZIF-8 (Qu@ZIF-8) nanoparticles were prepared through a one-step self-assembly technique. Specifically, 0.4 g of Zn(NO_3_)·6H_2_O and 2.76 g of 2-methylimidazole were each dissolved in 10 mL of deionized water to prepare the Zn(NO_3_)·6H_2_O solution and the 2-methylimidazole solution. The two solutions were then rapidly mixed and subjected to magnetic stirring for 5 min. The resulting mixture was subjected to centrifugation at 5000 rpm for 10 min to isolate the precipitate. This precipitate was then rinsed three times with deionized water and freeze-dried overnight at −80 °C to yield ZIF-8 nanoparticles. The synthesis of Qu@ZIF-8 nanoparticles followed a similar procedure, starting with the dissolution of 30 mg of quercetin in 1 mL of anhydrous ethanol. Next, the aforementioned 2-methylimidazole solution and the Zn(NO_3_)·6H_2_O solution were rapidly mixed with the quercetin solution to yield Qu@ZIF-8 nanoparticles.

Characterization of ZIF-8 and Qu@ZIF-8 Nanoparticles: The morphology of ZIF-8 and Qu@ZIF-8 nanoparticles was observed through scanning electron microscopy (SEM) (ZEISS GeminiSEM 300, ZEISS, Oberkochen, Germany). Five random SEM fields were selected for quantitative analysis of nanoparticle size distribution using ImageJ software (Version 1.54i). The chemical states of the nanoparticles were characterized through X-ray photoelectron spectroscopy (XPS) (Thermo Scientific K-Alpha, Thermo Fisher Scientific, Waltham, MA, USA). Additionally, the bulk structure of the nanoparticles was assessed through X-ray diffraction (XRD) (Bruker D8 Advance, Bruker, Billerica, MA, USA), with results compared to crystal simulation data provided by the Cambridge Crystallographic Data Centre (CCDC) (https://www.ccdc.cam.ac.uk, accessed on 24 October 2024).

Drug-Loading Efficiency Assessment: A mixture was prepared by dissolving 3 mg of Qu@ZIF-8 in 50 µL of hydrochloric acid, followed by the addition of anhydrous ethanol to reach a total volume of 2 mL. The absorbance at 370 nm was determined with a UV–Vis spectrophotometer (Thermo Scientific Evolution, Thermo Fisher Scientific, Waltham, MA, USA). The mass of the loaded drug was determined by utilizing a standard curve, and the drug-loading efficiency was determined with the formula provided below:$$Drug\;loading\;efficiency=\left(\frac{Mass\;of\;drug\;loaded}{Mass\;of\;nanoparticle}\right)\times 100\%.$$

Cell Culture: Mouse embryonic fibroblast cells (NIH-3T3) were cultured in DMEM media containing 10% calf serum and the recommended dosage of antibiotics (100 U/L penicillin and 100 mg/L streptomycin). Mouse mononuclear macrophage cells (RAW264.7) were cultured in DMEM media supplemented with 10% fetal bovine serum and the recommended dosage of antibiotics. Human Umbilical Vein Endothelial Cells (HUVECs) were cultured in DMEM/F12 media with 10% fetal bovine serum and the recommended dosage of antibiotics. All cell lines were passaged fewer than five times and incubated at 37 °C in a humidified atmosphere with 5% CO_2_.

In Vitro Cytotoxicity Assay: NIH-3T3 cells (1 × 10^4^ cells/well) were inoculated in a 96-well plate. Once the cells had adhered to the plate, the culture medium was substituted with media containing different concentrations of ZIF-8, quercetin, and Qu@ZIF-8. After a 24 h incubation, the CCK-8 assay was utilized to assess cell viability and determine the cytotoxic effects of ZIF-8, quercetin, and Qu@ZIF-8 on NIH-3T3 cells.

Preparation of GelMA: GelMA was synthesized following established methods [27]. Specifically, 5 g of gelatin was added into 50 mL of phosphate-buffered saline (PBS) until fully dissolved. At a temperature of 50 °C, 4 mL of methacrylic anhydride was introduced into the gelatin solution slowly (0.5 mL/min), and the mixture was stirred magnetically for 3 h. To terminate the reaction, 200 mL of pre-warmed PBS was added. The resulting mixture underwent continuous dialysis at 40 °C over a period of 7 days, with deionized water being replaced every 8 h. Following the dialysis process, the GelMA product was acquired through freeze-drying.

Characterization of GelMA: The chemical structure of GelMA was analyzed with proton nuclear magnetic resonance spectroscopy (^1^H-NMR) on a Bruker Avance 600 MHz spectrometer (Bruker, Billerica, MA, USA). The degree of functionalization of GelMA was evaluated through a previously reported method [28]. Briefly, 2,4,6-trinitrobenzenesulfonate (TNBS) solution was added to both GelMA and gelatin solutions, allowing for a complete reaction. The absorbance of the resulting samples was measured at 340 nm with a UV–Vis spectrophotometer. The degree of functionalization was then determined with the formula provided below:$$Degree\;of\;functionalization=\left(1-\frac{absorbance\;of\;GelMA}{absorbance\;of\;Gelatin}\right)\times 100\%.$$

Preparation of Drug-Loaded Hydrogel: To prepare the hydrogel precursor solution, a solution was prepared by dissolving 100 mg of GelMA and 3 mg of LAP in 1 mL of PBS. To facilitate drug loading, 80 nmol of Qu@ZIF-8 nanoparticles were then introduced into the precursor solution. After vortex mixing, the resulting solution was transferred into a polytetrafluoroethylene (PTFE) mold and crosslinked under UV light at 365 nm for 10 s to obtain the GelMA hydrogel containing 80 µM Qu@ZIF-8, designated as Qu@ZIF-8-Gel. Following the same preparation protocol, a pure GelMA hydrogel without any drug components was synthesized and labeled as Gel. Additionally, hydrogels containing 16 µM quercetin or 80 µM ZIF-8 were prepared, designated as Qu-Gel and ZIF-8-Gel, respectively.

Microscopic Characterization of Hydrogel Structure: The hydrogels’ microstructure was analyzed with a cryo-SEM (Titan Krios G3, FEI, Hillsboro, OR, USA). The hydrogels’ pore size and porosity were subsequently quantified with ImageJ (Version 1.54i).

Mechanical Evaluation of Hydrogels: The hydrogels’ rheological properties were assessed with a rheometer (Thermo Mars40, Thermo Fisher Scientific, Waltham, MA, USA), measuring both the storage modulus (G′) and the loss modulus (G″). Dynamic frequency sweeps were done across a frequency range from 0.1 to 10 Hz at a strain of 1%. The compression stress–strain curve of the hydrogel was determined using a universal testing machine (Instron 5969, Instron, Boston, MA, USA). The hydrogel was shaped into a cylinder with a height of 5 mm and a diameter of 10 mm, applying a load of 100 N at a loading rate of 0.5 mm/min.

Evaluation of Hydrogel Swelling Properties: Once freeze-drying was completed, the initial mass of the hydrogels was documented as M0. The hydrogels were subsequently submerged in PBS at 37 °C, with their weights being measured every 8 h. The mass at each time point was recorded as Mt. The swelling ratio at each time point was determined with the formula provided below:$$Swelling\;ratio=\left(\frac{Mt-M0}{M0}\right)\times 100\%.$$

Evaluation of Hydrogel Degradation Properties: After freeze-drying, the initial mass of the hydrogels was recorded as M0. The hydrogels were then submerged in PBS containing 0.2 U/mL type I collagenase at 37 °C. At designated time intervals, the hydrogels were freeze-dried and weighed, with the mass recorded as Mt. For samples that exhibited loss of integrity, the hydrogels were filtered through filter paper and freeze-dried. The degradation rate was determined with the formula provided below:$$Degradation\;rate=\left(\frac{M0-Mt}{M0}\right)\times 100\%.$$

Evaluation of Ion Release Capability: Hydrogel samples of ZIF-8-Gel and Qu@ZIF-8-Gel, each with a volume of 1 cm^3^, were prepared. The hydrogels were then treated with acidic PBS at pH 5.6 and alkaline PBS at pH 7.4 to assess their ion release capabilities in acidic and alkaline environments. For the acidic environment, the hydrogels were submerged in PBS, with the pH adjusted to 5.6 at 37 °C. At designated time intervals, the leachate was collected continuously, and the concentration of released zinc ions was assessed with inductively coupled plasma mass spectrometry (ICP-MS) (PerkinElmer NexION 300X, PerkinElmer, Waltham, MA, USA). The cumulative release rates of zinc ions were calculated based on the total mass of zinc in the samples.

Evaluation of Quercetin Release Capability: To assess quercetin release capability, Qu-Gel hydrogels with a volume of 1 cm^3^ containing 4.9 μg of quercetin and Qu@ZIF-8-Gel hydrogels containing 23.4 μg of Qu@ZIF-8 were prepared. Following the methodology used in the zinc ion release experiments, leachate (anhydrous ethanol) was collected at specified time points. The concentration of the test solution was adjusted to a reasonable range by using the evaporation method. The amount of quercetin released was measured with a UV–Vis spectrophotometer. The cumulative release rates of quercetin were calculated.

Biocompatibility Assessment of Hydrogels: NIH-3T3 cells (1 × 10^4^ cells/well) were inoculated in a 96-well plate. The cell samples were treated with the leachates from the various pre-prepared hydrogels, and after 24 h, the influence of the hydrogels on cell proliferation was analyzed via the CCK-8 assay. To further validate biocompatibility, a co-culture system of hydrogels and NIH-3T3 cells was established using Transwell chambers. Specifically, NIH-3T3 cells (1 × 10^5^ cells/well) were inoculated in a 24-well plate, and precursor solutions of the hydrogels were introduced into the Transwell chambers. After UV light exposure to induce gelation, the chambers were placed in the 24-well plate for co-culture with NIH-3T3 cells for 24 h. The biocompatibility of each hydrogel was evaluated through the CCK-8 assay and Calcein-AM/PI staining. ImageJ software (Version 1.54i) was employed for quantitative analysis of live and dead cell counts based on the Calcein-AM/PI staining results.

Scratch Assay: The influence of hydrogels on cell migration was investigated via a scratch assay. NIH-3T3 cells (1 × 10^6^ cells/well) were inoculated in a six-well plate. Once the cells reached confluence, a scratch was made with a 200 µL pipette tip, and images were captured under a microscope. Subsequently, the cells were co-cultured with the hydrogels for 24 h, after which scratch images were collected again. The wound-healing rate was analyzed and calculated using ImageJ software (Version 1.54i).

Cytoskeletal Staining: The influence of hydrogels on cell morphology was investigated with cytoskeletal staining. Cell culture slides were positioned in a 6-well plate, and NIH-3T3 cells (1 × 10^5^ cells/well) were inoculated onto the slides. After co-culturing with the hydrogels in Transwell chambers for 24 h, cytoskeletal staining was performed to evaluate cellular morphology.

Evaluation of Cell Adhesion on Hydrogel Surfaces: 200 µL of each hydrogel precursor solution was introduced into a 24-well plate. After allowing the hydrogels to evenly spread and solidify under UV light, NIH-3T3 cells (1 × 10^5^ cells/well) were inoculated onto the surface of the hydrogels. Following a 24 h incubation, the wells were rinsed three times with PBS to eliminate non-adherent cells. Cell adhesion status on the surfaces of the hydrogels was assessed using Calcein-AM/PI staining.

RT-qPCR Detection of the Immunomodulatory Effects of Hydrogels: RAW264.7 cells (1 × 10^6^ cells/well) were inoculated in a 6-well plate. After co-culturing with the hydrogels in Transwell chambers for 72 h, total RNA extraction, reverse transcription, and cDNA amplification were made according to standard procedures. PCR detection was conducted via a real-time fluorescence quantitative PCR system (Thermo QuantStudio 3, Thermo Fisher Scientific, Waltham, MA, USA). The data were processed through the 2-ΔΔCt approach. The sequences of the primers are detailed in Table 1.

Immunofluorescence Staining to Assess the Immunomodulatory Effects of Hydrogels: RAW264.7 cells (1 × 10^5^ cells/well) were inoculated on the surface of cell culture slides within a six-well plate. Following co-culture with the hydrogels in Transwell chambers for 72 h, the cells were fixed, permeabilized, and blocked according to the standard protocol. Primary antibodies targeting CD68, CD86, and CD206 were applied and kept on ice for 24 h. Afterward, the samples were incubated with fluorescently labeled secondary antibodies at 37 °C for 60 min, followed by a 30 s counterstaining with DAPI. The images were obtained via a fluorescence microscope.

ELISA Detection of the Immunomodulatory Effects of Hydrogels: RAW264.7 cells (1 × 10^5^ cells/well) were inoculated in a 24-well plate. After co-culturing with the hydrogels in Transwell chambers for 72 h, the levels of interleukin-10 (IL-10) and interleukin-12 (IL-12) in the supernatants were measured via an ELISA kit, following the manufacturer’s guidelines.

RT-qPCR Detection of the Angiogenic Effects of Hydrogels: HUVEC cells (1 × 10^6^ cells/well) were inoculated in a 6-well plate. RT-qPCR was performed to assess the levels of angiogenesis-related genes. The sequences of the primers are detailed in Table 2.

Immunofluorescence Staining for the Angiogenic Effects of Hydrogels: HUVEC cells (1 × 10^6^ cells/well) were inoculated on cell culture slides within a six-well plate. After co-culturing with the hydrogels in Transwell chambers for 72 h, immunofluorescence staining for the vascular endothelial growth factor (VEGF) marker was conducted following the previously described procedures.

Tube Formation Assay: The hydrogels’ angiogenic capability was assessed using a tube formation assay. Initially, we dispensed 100 µL of Matrigel matrix into each well of a 24-well plate and allowed it to gel at 37 °C. Subsequently, HUVEC cells (5 × 10^5^ cells/well) were inoculated on the surface of the gel and co-cultured with the hydrogels for 6 h. The parameters of capillary length and branch points were quantitatively analyzed using ImageJ software (Version 1.54i).

Assessment of Antibacterial Properties of Hydrogels: The hydrogels’ antibacterial properties were assessed using the plate colony counting method. First, 1 mL of each hydrogel precursor solution was introduced into a 24-well plate and allowed to solidify under UV light. Next, 100 µL of bacterial suspension (1 × 10^6^ CFU/mL) containing either *Staphylococcus aureus* (*S. aureus*) or *Escherichia coli* (*E. coli*) was added to the surface of the gels and incubated at 37 °C for 24 h. After incubation, 10 µL of the bacterial suspension was diluted 100-fold, and 40 µL of this dilution was spread onto Luria-Bertani (LB) agar plates. The plates were then incubated at 37 °C for 12 h, after which the colonies were counted. The antibacterial efficiency was determined with the formula provided below:$$Antimicrobial\;efficiency=\left(1-\frac{treatment\;group\;colony\;count}{control\;group\;colony\;count}\right)\times 100\%.$$

To explore the antibacterial performance and mechanism of the hydrogels, 10 µL of *S. aureus* or *E. coli* bacterial suspension was added onto sterile filter paper discs, which were then positioned on the pre-gelled hydrogels. Subsequently, 100 µL of LB broth was introduced into the system, and the samples were incubated at 37 °C for 24 h. After incubation, the samples underwent standard SEM preparation procedures, including fixation, dehydration, and critical point drying, allowing for the observation of bacterial morphological changes using SEM.

Skin Wound Model: The complete protocol for animal experiments was approved by the Ethics Committee for Experimental Animals of West China Hospital, Sichuan University (2024-0920010). Twenty-four six-week-old Sprague-Dawley (SD) rats were anesthetized via intraperitoneal injection of sodium pentobarbital (50 mg/kg) and randomly assigned to four groups. After disinfecting the skin, a full-thickness circular skin defect measuring 1 cm in diameter was established on the dorsal region of each rat. Subsequently, the precursor solutions of the hydrogels were injected into the wound area, and the gels were formed under UV light exposure. The wound-healing progression was documented at specified time points.

Histological Staining: On day 14, three rats from each group were selected for histological and immunohistochemical analysis. Skin wound specimens were harvested and submerged in 4% paraformaldehyde for 24 h for fixation. The specimens were subsequently processed into 5 µm thick skin slices and stained with hematoxylin-eosin (H&E) to compare skin regeneration among different hydrogel groups. Additionally, immunohistochemical staining was performed to assess the deposition of type I and type III collagen, as well as the expression levels of CD86 and CD206 in the tissues. To further elucidate the presence of neovascularization within the tissue, immunofluorescence staining was conducted to label regions positive for CD31 and α-smooth muscle actin (α-SMA).

Data Statistical Analysis: Statistical analysis was executed with SPSS 20.0 software. Quantitative data were presented as mean ± standard deviation (Mean ± SD). Samples were compared through one-way analysis of variance (ANOVA), with a significance criterion of *p* < 0.05.

## 3. Results

### 3.1. Synthesis and Characterization of Qu@ZIF-8

In this study, ZIF-8 and Qu@ZIF-8 nanoparticles were successfully synthesized via a one-step self-assembly method. SEM images, as illustrated in Figure 1B, demonstrated that both ZIF-8 and Qu@ZIF-8 had uniform cubic structures, with average particle sizes of 75.0 ± 1.9 nm for ZIF-8 and 75.8 ± 1.3 nm for Qu@ZIF-8. To assess the impact of quercetin incorporation on the crystal structure of ZIF-8, we performed XRD analysis on both materials. The results revealed that Qu@ZIF-8 retained diffraction peaks consistent with those of ZIF-8 (see Figure 1A), suggesting that quercetin loading does not disrupt the crystal structure of ZIF-8.

Furthermore, we characterized the elemental composition and valence states of ZIF-8 and Qu@ZIF-8 using XPS. The C 1s spectrum of ZIF-8 was deconvoluted into two peaks at 284.8 eV and 285.6 eV, corresponding to C-C and C-N bonds, respectively. In contrast, the C 1s spectrum of Qu@ZIF-8 exhibited three additional peaks at 286.0 eV, 288.1 eV, and 291.9 eV, where the peaks at 286.0 eV and 288.1 eV corresponded to C-O and C=O bonds present in the quercetin structure. Additionally, the aromatic ring structure in quercetin allowed for the observation of a satellite peak at 291.9 eV (see Figure 1C,D). These results confirm the successful incorporation of quercetin into ZIF-8.

Subsequently, we assessed the drug-loading efficiency of quercetin onto ZIF-8 using UV–Vis spectrophotometry, which yielded a drug-loading efficiency of 20.9%.

### 3.2. Synthesis and Characterization of GelMA

In this study, we synthesized GelMA by reacting gelatin with methacrylic anhydride. To confirm the successful grafting of methacryloyl groups onto the gelatin backbone, we performed ^1^H-NMR spectroscopy on the synthesized samples. The results, as shown in Figure 2A, revealed that the peaks labeled a and b in the range of 5–6 ppm correspond to the protons of the methacryloyl groups attached to the lysine and hydroxylysine residues in the gelatin framework. Additionally, the peak labeled d near 1.8 ppm corresponds to the methyl protons within the methacryloyl group. A significant reduction in the methylene proton peak around 2.9 ppm was also observed in the GelMA spectrum compared to gelatin, indicating the successful synthesis of GelMA.

Furthermore, we assessed the degree of functionalization of GelMA using the TNBS assay, which demonstrated that approximately 72.6 ± 4.8% of the amine groups in gelatin were functionalized with methacrylamide groups (see Figure 2B).

### 3.3. Synthesis and Characterization of Qu@ZIF-8-Gel

Based on the established safe concentration ranges of the drugs (quercetin, ZIF-8, and Qu@ZIF-8), we prepared precursor solutions for each hydrogel group. These precursor solutions were then subjected to UV light to achieve crosslinking, resulting in drug-loaded hydrogels. The macroscopic images of the hydrogels before and after gelation are shown in Figure 2C; all hydrogels could rapidly gel under UV light, suggesting that the incorporation of the drugs does not adversely affect the gelation performance of GelMA hydrogels.

To further assess whether the drugs impact the microstructure of the hydrogels, we employed cryo-SEM for characterization. As depicted in Figure 2D, all hydrogels exhibited a porous structure with interconnected pores uniformly distributed throughout the hydrogel matrix. Subsequent quantitative analysis using ImageJ revealed no significant differences in pore size and porosity among the different hydrogels (see Figure 3A,B), suggesting that the microstructure of the hydrogels remains unchanged with the addition of the drugs.

We then compared the mechanical properties, degradation behavior, and swelling characteristics of the various hydrogels. The rheological measurements indicated that both G′ and G″ of the hydrogels were comparable across groups, with G′ significantly exceeding G″ (see Figure 3C), confirming that the incorporation of the drugs does not affect the mechanical properties of the hydrogels, and all hydrogels are in a solid state. The compression experiments further validated this result, indicating that the compression Young’s modulus of each hydrogel is comparable, approximately 8.5 kPa.

The swelling experiments demonstrated that all hydrogels reached swelling equilibrium after 56 h, with similar swelling rates of approximately 800% at equilibrium (see Figure 3F). This suggests that the introduction of the drugs does not negatively impact the swelling properties of the hydrogels. Furthermore, the favorable swelling characteristics of these hydrogels enable them to absorb exudate from wounds while maintaining a moist environment.

Degradation studies showed that the incorporation of drugs did not affect the degradation performance of the hydrogels, with all formulations completely degrading within 24 days (see Figure 3G). This ensures that the hydrogels will not interfere with the wound-healing process.

Subsequently, we evaluated the release profiles of quercetin and zinc ions from the drug-loaded hydrogels. The results, as illustrated in Figure 3H,I, indicated that encapsulating quercetin within ZIF-8 significantly reduced its release rate. On day 21, the cumulative release rates of quercetin from the Qu@ZIF-8-Gel hydrogel under different pH environments were 18.9% (pH = 5.6) and 16.6% (pH = 7.4), respectively. Notably, under alkaline conditions, the release rate of quercetin from the Qu-Gel hydrogel was significantly accelerated, with a cumulative release rate of 40.6% on day 21, which was much higher than the 26.5% observed under acidic conditions. However, the release of quercetin from the Qu@ZIF-8-Gel hydrogel was found to be minimally affected by changes in environmental pH.

### 3.4. Evaluation of Cytotoxicity of Quercetin, ZIF-8, and Qu@ZIF-8

To determine the safe concentration ranges for quercetin, ZIF-8, and Qu@ZIF-8, we co-cultured NIH-3T3 cells with varying concentrations of these substances. The effects on cell proliferation were evaluated with the CCK-8 assay. The findings showed that when the concentrations of ZIF-8 and Qu@ZIF-8 exceeded 80 µM, a notable decrease in the proliferation of NIH-3T3 cells was observed, suggesting that the safe concentration range for both ZIF-8 and Qu@ZIF-8 is 0–80 µM. Additionally, quercetin concentrations above 16 µM resulted in a notable reduction in cell proliferation, indicating that the safe concentration range for quercetin is 0–16 µM (see Figure 4A). Therefore, we selected 16 µM of quercetin and 80 µM of ZIF-8 and Qu@ZIF-8 for subsequent experimental studies.

### 3.5. Biocompatibility Evaluation of Qu@ZIF-8-Gel

We initially assessed how the drug-loaded hydrogels affect cell proliferation with the CCK-8 assay. Results showed no significant changes in absorbance at 450 nm for the drug-loaded hydrogels, suggesting that they do not adversely affect cell proliferation. To further validate the biocompatibility of the drug-loaded hydrogels, we employed the Calcein-AM/PI staining method to evaluate cell viability in each sample. Fluorescence images and quantitative analysis revealed that the drug-loaded hydrogels did not notably change the counts of live and dead cells (see Figure 4C,F), further confirming their non-cytotoxicity.

Subsequently, cytoskeletal staining results demonstrated that cells in all groups exhibited morphological characteristics consistent with NIH-3T3 cells, indicating that the drug-loaded hydrogels do not significantly impact cell morphology. We also investigated cell migration using a scratch assay, which found that the Qu@ZIF-8-Gel group significantly accelerated the wound-healing process (see Figure 4E,G), suggesting that The Qu@ZIF-8-Gel hydrogel can notably enhance cell migration ability.

Finally, NIH-3T3 cells were inoculated onto the surface of the drug-loaded hydrogels to evaluate their effect on cell adhesion. The results indicated that NIH-3T3 cells successfully adhered to and proliferated on the surface of the drug-loaded hydrogels (see Figure 4I), further demonstrating that the drug-loaded hydrogels do not significantly affect cell adhesion.

### 3.6. Evaluation of the Immunomodulatory Effects of Qu@ZIF-8-Gel

Currently, we first assessed the levels of the inducible nitric oxide synthase (iNOS) gene associated with M1 polarization and the CD206 gene associated with M2 polarization using RT-qPCR. The results demonstrated that both the Qu-Gel and Qu@ZIF-8-Gel hydrogels significantly upregulated CD206 expression while downregulating iNOS expression (see Figure 5B). To illustrate these results further, we employed immunofluorescence staining to confirm the expression of macrophage polarization markers. Fluorescent images revealed that the Qu@ZIF-8-Gel hydrogel significantly upregulated the M2 macrophage marker CD206 while downregulating the M1 macrophage marker CD86 (see Figure 5C).

Subsequently, we determined the amounts of anti-inflammatory cytokine IL-10 and pro-inflammatory cytokine IL-12 in the culture supernatant. The results, as shown in Figure 5A, indicated that the ZIF-8-Gel, Qu-Gel, and Qu@ZIF-8-Gel hydrogels significantly upregulated the levels of IL-10 while downregulating the levels of IL-12. Furthermore, compared to the ZIF-8-Gel and Qu-Gel hydrogels, the Qu@ZIF-8-Gel hydrogel had a more pronounced effect on the expression of IL-10 and IL-12 in RAW264.7 cells.

### 3.7. Evaluation of the Angiogenic Effects of Qu@ZIF-8-Gel

Currently, we first detected the levels of angiogenesis-related genes, specifically VEGF and HIF-1α, using RT-qPCR. HIF-1α is a critical regulator of VEGF that can upregulate its expression in response to hypoxic conditions, thereby promoting endothelial cell proliferation and migration. The RT-qPCR results indicated that both the ZIF-8-Gel and Qu@ZIF-8-Gel hydrogels significantly upregulated the levels of VEGF and HIF-1α in HUVEC cells (see Figure 5E). To further validate these findings at the protein level, we employed immunofluorescence staining to evaluate VEGF expression. The findings demonstrated that the fluorescence intensity of VEGF in samples from the ZIF-8-Gel and Qu@ZIF-8-Gel groups was markedly greater compared to the other groups (see Figure 5D), suggesting a significant upregulation in VEGF expression.

Subsequently, we conducted tube formation assays to evaluate the angiogenic potential of the various hydrogels. The results revealed that samples from the ZIF-8-Gel and Qu@ZIF-8-Gel groups exhibited denser tubular structures, with significantly more branch points compared to other groups. Furthermore, the capillary length in the Qu@ZIF-8-Gel group was notably longer than that in the control and Qu-Gel groups (see Figure 5F,G).

### 3.8. Evaluation of the Antibacterial Effects of Qu@ZIF-8-Gel

Initially, we employed the plate colony counting method to validate the antibacterial capabilities of the hydrogels. The results indicated that the ZIF-8-Gel, Qu-Gel, and Qu@ZIF-8-Gel hydrogels significantly inhibited the growth of *S. aureus* or *E. coli* (see Figure 6B). Among these, the Qu@ZIF-8-Gel hydrogel exhibited the most pronounced antibacterial effect, achieving an antibacterial efficiency of 99.4 ± 0.3% against *E. coli* and 98.8 ± 0.6% against *S. aureus* (see Figure 6C).

Subsequently, we utilized SEM to examine the morphological changes in bacteria treated with different hydrogels. As illustrated in Figure 6D, morphological alterations were evident in the bacteria treated with the ZIF-8-Gel, Qu-Gel, and Qu@ZIF-8-Gel hydrogels. The bacteria aggregated into irregular clusters, with pronounced surface shrinkage and loss of normal morphological characteristics.

### 3.9. In Vivo Evaluation of Drug-Loaded Hydrogels on Wound Healing

In vitro studies have indicated that the Qu@ZIF-8-Gel hydrogel is a strong candidate for wound dressings in treating skin injuries. Therefore, we established a skin wound model in SD rats to explore the wound-healing-promoting effects of the Qu@ZIF-8-Gel hydrogel. Macroscopic observations revealed that the wound-healing rate in rats receiving drug-loaded hydrogels was significantly faster compared to those treated with plain hydrogel (see Figure 7B). Specifically, the wound-healing rates for the Qu@ZIF-8-Gel group reached 65.9% and 99.5% on days 7 and 14, respectively, while the Gel group showed only 30.5% and 69.3% healing rates on the same time points (see Figure 7C). Histological analysis via H&E staining corroborated these findings, showing that after 14 days, wounds in the Qu@ZIF-8-Gel group were nearly completely healed, with dense granulation tissue and newly formed epidermis and dermis observed. In contrast, the other groups displayed incomplete healing, with discontinuous epidermal and dermal structures still evident (see Figure 7D).

In the subsequent in vivo experiments, we aimed to further validate the angiogenic effects of the Qu@ZIF-8-Gel hydrogel by performing immunofluorescence analysis to detect the CD31 and α-SMA in tissue samples. CD31 is a cell adhesion molecule commonly used as a marker for endothelial cells, while α-SMA is a smooth muscle-specific actin that serves as a marker for vascular smooth muscle cells [29]. The results of the immunofluorescence analysis showed a notable rise in CD31 and α-SMA-positive lumen-like structures in samples treated with the ZIF-8-Gel and Qu@ZIF-8-Gel hydrogels (see Figure 7E).

To further investigate whether the improvement in skin wound healing is associated with macrophage polarization, we performed immunohistochemical staining on tissue samples. The findings indicated that the drug-loaded hydrogels markedly improved the deposition of type I and type III collagen in the skin wounds. Additionally, the drug-loaded hydrogels markedly reduced the polarization of CD86^+^ M1 macrophages in the tissue while enhancing the polarization of CD206^+^ M2 macrophages (see Figure 7F,G).

## 4. Discussion

Rapid healing of skin wounds has consistently been a significant challenge in clinical practice. Typically, prolonged inflammation and bacterial infection during the wound-healing process can lead to tissue damage, impede angiogenesis, and result in delayed wound healing [4]. Therefore, the development of multifunctional wound dressings is urgent.

Hydrogels are biomedical polymers widely used in drug delivery, implants, and tissue engineering. Due to their excellent biocompatibility and ability to maintain a moist healing environment, hydrogels are employed to promote skin tissue repair and regeneration. For instance, Mariello and his research team developed a wound dressing with excellent stretchability and compressive properties by crosslinking polyvinyl alcohol/carboxymethyl cellulose and sericin through a freeze–thaw method, demonstrating its advantages in accelerating wound healing in in vitro experiments [30]. Furthermore, encapsulating bioactive drugs within hydrogels is also considered a practical approach for wound dressing preparation. For example, Zhang et al. incorporated polydopamine into an injectable hydrogel framework formed from oxidized sodium alginate and polyvinyl alcohol-bearing styrylpyridinium group (PVA-SBQ), leveraging the photothermal effect of polydopamine particles to exert antibacterial properties, which significantly accelerated the healing process of infected wounds [31]. Inspired by these studies, in the present research, we encapsulated quercetin-loaded ZIF-8 within the GelMA hydrogel to develop an injectable, photocrosslinked hydrogel dressing that possesses comprehensive activities, including immunomodulation, antibacterial properties, and pro-angiogenic effects, thereby facilitating the healing process of skin wounds.

In the current study, we successfully synthesized the GelMA and Qu@ZIF-8. Subsequent validation experiments confirmed that the degree of functionalization of GelMA and the loading efficiency of quercetin onto Qu@ZIF-8 were comparable to those reported in previous studies, thereby demonstrating the effectiveness of the synthesis method. Next, we determined the safe concentration of Qu@ZIF-8 and constructed the Qu@ZIF-8-Gel hydrogel at this concentration. A series of characterizations were performed, and the results confirmed that the incorporation of Qu@ZIF-8 did not adversely affect the original physicochemical properties of the GelMA hydrogel. Notably, the release rate of quercetin from the Qu-Gel hydrogel was markedly accelerated in alkaline conditions. This is likely due to the ionization of carboxyl groups in GelMA at alkaline pH, which leads to more pronounced swelling of the hydrogel [32]. Furthermore, we observed that the drug release from the Qu@ZIF-8-Gel hydrogel was minimally affected by changes in environmental pH. This phenomenon might be connected to the pH-responsive properties of ZIF-8, which maintains high stability under alkaline conditions [33], thereby ensuring a slow and sustained release of both quercetin and zinc ions. These characteristics enable the Qu@ZIF-8-Gel hydrogel to provide a more stable drug release profile that is not affected by fluctuations in the pH of the wound environment. We subsequently evaluated the biocompatibility of the Qu@ZIF-8-Gel hydrogel through in vitro experiments. The results confirmed that the Qu@ZIF-8-Gel hydrogel exhibited excellent biocompatibility, providing an appropriate microenvironment for cell proliferation, migration, and adhesion.

Wound healing is a multifaceted process that encompasses three overlapping phases: inflammation, proliferation, and remodeling [34]. When the inflammatory phase is prolonged due to an excessive inflammatory response, the healing process can be delayed, primarily driven by macrophage polarization [35]. Given the remarkable anti-inflammatory properties of ZIF-8 and quercetin, this research investigated the immunomodulatory effects of the drug-loaded hydrogels, with a particular focus on their impact on macrophage polarization. Macrophages can generally be classified into M1 and M2 types according to their activation status and functional roles. M1 macrophages are primarily involved in the initial stages of acute inflammation, triggering and amplifying local inflammatory responses, as well as clearing pathogens and toxic cellular debris from damaged tissues [36,37]. However, the persistent presence of M1 macrophages may lead to secondary tissue damage or chronic inflammation, thereby hindering the repair process [38]. 

In contrast, M2 macrophages, termed pro-repair macrophages, are crucial in the later stages of the inflammatory response by exerting anti-inflammatory effects and promoting cell proliferation [39], thus facilitating the resolution of inflammation and healing of wounds. In a previous study, Zhou et al. developed a double-network hydrogel biomaterial composed of snail glycosaminoglycan and GelMA for the treatment of diabetic wounds. The experimental results demonstrated that this biomaterial significantly alleviated inflammation while promoting M2 polarization of macrophages, effectively transitioning diabetic wounds from the inflammatory stage to the proliferative stage, ultimately achieving wound healing [40]. In this study, we observed that ZIF-8-Gel, Qu-Gel, and Qu@ZIF-8-Gel hydrogels upregulated the expression of IL-10 and CD206, which are closely associated with M2 macrophage polarization while downregulating the expression of IL-12 and iNOS, which are related to M1 macrophage polarization. Furthermore, compared to the ZIF-8-Gel and Qu-Gel hydrogels, the Qu@ZIF-8-Gel hydrogel exhibited a more pronounced effect on the expression of the aforementioned genes or proteins, with IL-10 levels increasing to 116.7 ± 18.9 pg/mL and the relative expression of the CD206 gene rising to 3.2 ± 0.9, significantly higher than other groups. 

Conversely, the expression of IL-12 decreased to 28.4 ± 3.0 pg/mL, and the relative expression of the iNOS gene also decreased to 0.5 ± 0.1, markedly lower than the other groups (Figure 5A,B). Subsequent immunofluorescence detection (Figure 5C) and immunohistochemical staining of tissue samples (Figure 7F) also revealed the same trend. These results suggest that the zinc ions and quercetin components in the Qu@ZIF-8-Gel hydrogel can synergistically promote macrophage polarization toward the M2 phenotype while inhibiting the M1 phenotype, thereby exhibiting more significant immunomodulatory effects. In light of previous research findings, we believe that this regulatory effect on the local inflammatory environment will be beneficial for the rapid healing of skin wounds. Furthermore, subsequent in vivo experiments confirmed our hypothesis, showing that the skin wound healing rate in rats implanted with the Qu@ZIF-8-Gel hydrogel significantly accelerated. By day 14 post-implantation, the skin wounds in rats had nearly completely healed, and H&E staining results indicated that the newly formed epidermal tissue had fully covered the wound site (Figure 7B,D), confirming the potential application of the Qu@ZIF-8-Gel hydrogel as a wound dressing.

During the proliferation phase of wound healing, a key accompanying activity is angiogenesis. The formation of new blood vessels within the wound tissue ensures that nutrients and oxygen are delivered to the tissue in a balanced manner while waste products and carbon dioxide are removed in a timely fashion [41]. Previous research evidence indicates that angiogenesis is crucial for the healing of skin wounds. For example, Guan et al. proposed a sustained oxygenation system composed of oxygen-release microspheres and reactive oxygen species (ROS)-scavenging hydrogel for skin wound treatment. Experimental results confirmed that sustained oxygenation enhanced the expression of growth factors such as vascular endothelial growth factor-A (VEGFA) and platelet-derived growth factor-B (PDGFB) in skin cells, thereby promoting angiogenesis in wounds and ultimately improving wound-healing rates [42]. In this study, an important issue to note is that when the concentration of quercetin exceeds 50 μM, it significantly inhibits the expression of vascular endothelial growth factor receptor 2 (VEGFR2) in human umbilical vein endothelial cells (HUVECs) and suppresses tubular structure formation [43], thereby exhibiting anti-angiogenic properties. To address this issue, the present study encapsulated quercetin within ZIF-8, achieving a cumulative release of 16.6 ± 2.1% by day 21 in an environment with a pH of 7.4 (Figure 3H), thus successfully facilitating its sustained release. Additionally, due to the pro-angiogenic effects of zinc ions, the ZIF-8-Gel hydrogel significantly upregulated the expression of VEGF and HIF-1α genes in HUVECs, substantially promoting the formation of vascular networks. Notably, compared to the control group, the Qu@ZIF-8-Gel hydrogel also significantly upregulated the expression of VEGF and HIF-1α, with relative expression levels of VEGF at 3.6 ± 0.5 and HIF-1α at 3.3 ± 0.8 (Figure 5E). The capillary length in tube formation assays was also significantly increased, with the Qu@ZIF-8-Gel group showing a length of 21.4 ± 2.8 mm, much higher than the 12.8 ± 3.0 mm observed in the control group (Figure 5F,G). Subsequent VEGF immunofluorescence results further validated these findings (Figure 5D), further confirming that the system effectively mitigated the anti-angiogenic effects of quercetin while harnessing the pro-angiogenic properties of ZIF-8. In subsequent in vivo experiments, the expression of angiogenic markers CD31 and α-SMA in rat skin treated with the Qu@ZIF-8-Gel and ZIF-8-Gel hydrogels was significantly increased (Figure 7E), further elucidating the pro-angiogenic effect of this hydrogel.

As an important interface between the body and the surrounding environment, the skin hosts a diverse array of microbial communities [44]. However, when the skin is damaged, it provides an opportunity for harmful bacteria to invade living tissues, leading to wound infections and even severe tissue damage [45]. Quercetin exhibits a wide range of antibacterial properties by compromising the integrity of bacterial membrane structures, inhibiting nucleic acid synthesis, and preventing biofilm formation [18], making it a strong candidate for novel antibacterial agents. ZIF-8 is considered a typical metal–organic framework (MOF) with antibacterial properties; it can disrupt bacterial membrane structures and cause the leakage of cytoplasmic contents through the release of zinc ions [46]. Due to the good antibacterial performance of ZIF-8, it is often used as a novel antibacterial agent in the research of infectious disease treatments. For example, Liu et al. encapsulated ZIF-8 in the GelMA hydrogel to construct an antibacterial hydrogel system for the treatment of periodontitis. The results showed that this hydrogel significantly reduced bacterial load and exhibited great potential in treating periodontitis [47].

In this study, we encapsulated quercetin within ZIF-8 to achieve enhanced antibacterial effects. The colony-forming unit (CFU) count results showed that compared to the control group and other hydrogel groups, the Qu@ZIF-8-Gel hydrogel effectively inhibited the proliferation of *S. aureus* and *E. coli*, achieving antibacterial efficiencies of 99.4 ± 0.3% against *E. coli* and 98.8 ± 0.6% against *S. aureus* (Figure 6B,C). This confirms that the Qu@ZIF-8-Gel hydrogel successfully integrates the antibacterial effects of quercetin and ZIF-8, demonstrating superior antibacterial performance. Subsequently, SEM results confirmed that the Qu@ZIF-8-Gel hydrogel caused damage to the bacterial membrane structure (Figure 6D), further validating previous research findings that both quercetin and ZIF-8 can exert their antibacterial effects by disrupting the integrity of bacterial membrane structures.

Currently, the specific mechanisms by which the Qu@ZIF-8-Gel hydrogel promotes skin wound healing remain limited and require further investigation to elucidate its mode of action. Additionally, our present study has focused solely on the application of the Qu@ZIF-8-Gel hydrogel in conventional wounds. Future research will explore the healing-promoting effects of this hydrogel in specialized wound types, such as those associated with diabetes.

## 5. Conclusions

A novel GelMA-based drug-loaded hydrogel with immunomodulatory, antibacterial, and pro-angiogenic properties has been developed as an innovative dressing material to facilitate the rapid healing of skin wounds. This hydrogel dressing is made up of photo-crosslinked GelMA and quercetin-loaded ZIF-8. Upon injection into the wound site, GelMA undergoes photo-crosslinking to form a gel, demonstrating excellent injectability. Experimental results confirm that the expression levels of M2 polarization-related genes and proteins in the Qu@ZIF-8-Gel hydrogel group were significantly higher than those in the control group, indicating its capability to effectively stimulate M2 macrophage polarization. Additionally, the Qu@ZIF-8-Gel hydrogel significantly inhibited the proliferation of *S. aureus* and *E. coli*, achieving an antibacterial efficiency of 99.4 ± 0.3% against *E. coli* and 98.8 ± 0.6% against *S. aureus*. Furthermore, the expression levels of angiogenesis-related genes and proteins in the Qu@ZIF-8-Gel hydrogel group were notably increased compared to the control group, and in tube formation assays, the vascular length nearly doubled compared to the control group, suggesting significant angiogenic effects. Subsequent animal studies further validated the immunomodulatory and angiogenic capabilities of the Qu@ZIF-8-Gel hydrogel, significantly promoting the repair process of skin wounds. Notably, this hydrogel also exhibits good biocompatibility and ease of synthesis. In summary, the Qu@ZIF-8-Gel hydrogel shows promising potential as a novel dressing material suitable for skin wound repair.

## Data Availability

The original contributions presented in the study are included in the article, further inquiries can be directed to the corresponding authors.

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
