# Peer review of "Injectable Photocrosslinked Hydrogel Dressing Encapsulating Quercetin-Loaded Zeolitic Imidazolate Framework-8 for Skin Wound Healing"

_pharmaceutics, 2024, doi:10.3390/pharmaceutics16111429_

Round 1
Reviewer 1 Report
Comments and Suggestions for Authors
Please see the attachment

Author Response
Comments 1: The title of the manuscript is too long.
Response 1: Thank you for pointing this out. We agree with this comment. Therefore, we have changed the title of the manuscript to “Injectable Photocrosslinked Hydrogel Dressing Encapsulating Quercetin-Loaded Zeolitic Imidazolate Framework-8 for Skin Wound Healing”.
Comments 2: In the introduction the authors should describe with more details the current methods for wound healing.
Response 2: Thank you for pointing this out. We agree with this comment. Therefore, we have added information regarding the current treatments for wound healing in the introduction section (lines 39-42), as follows:
Currently, treatment for skin wounds includes surgical debridement, grafting, dressing changes, and hyperbaric oxygen therapy, and more [4]. Among these, dressing changes have become the primary strategy for wound treatment due to their low cost and ease of application. However, in clinical practice, traditional dressings often struggle to sustain the moist conditions essential for optimal wound healing [6]. Additionally, changing dressings can cause secondary trauma to newly formed granulation tissue, adversely affecting healing efficiency and outcomes, while significantly increasing the workload of healthcare professionals [7]. Thus, there is a pressing need for a wound treatment strategy that balances simplicity and high effectiveness.
Comments 3: When discussing hydrogels as valuable tools for wound healing, the authors should add more references and describe briefly the types of hydrogels that are generally used for this purpose (e.g., https://doi.org/10.3390/gels10060412).
Response 3: Thank you for pointing this out. We agree with this comment. Therefore, we have added references and an explanation to the application of hydrogels in this field in the discussion section (lines 574-584), as follows:
For instance, Mariello and his research team developed a wound dressing with excellent stretchability and compressive properties by crosslinking polyvinyl alcohol/carboxymethyl cellulose and sericin through a freeze–thawing method, demonstrating its advantages in accelerating wound healing in in vitro experiments [30]. Furthermore, encapsulating bioactive drugs within hydrogels is also considered a practical ap-proach for wound dressing preparation. For example, Zhang et al. incorporated polydopamine into an injectable hydrogel framework formed from oxidized sodium alginate and polyvinyl alcohol bearing styrylpyridinium group (PVA-SBQ), leveraging the photothermal effect of polydopamine particles to exert antibacterial properties, which sig-nificantly accelerated the healing process of infected wounds [31]. Inspired by these studies, in the present research, we encapsulated quercetin-loaded ZIF-8 within GelMA hydrogel to develop an injectable photocrosslinked hydrogel dressing that possesses comprehensive activities, including immunomodulation, antibacterial properties, and pro-angiogenic effects, thereby facilitating the healing process of skin wounds.
Comments 4: In Scheme 1, are the drawings prepared with BioRender? If yes, the proper citation should be added in the caption.
Response 4: Thank you for pointing this out. We agree with this comment. Therefore, we have added a citation in the caption of Scheme 1.
Comments 5: Figure 1: the insets in B are too small, as well as the legends in G-K of Figure 2, and the axis categories in C,G of Figure 6.
Response 5: Thank you for pointing this out. We agree with this comment. Therefore, we have modified the above image as per your suggestion to make it clearer.
Comments 6: What is the mechanical rigidity of the proposed hydrogels (in compression and tension/bending mode)?
Response 6: Thank you for pointing this out. We agree with this comment. Therefore, we have supplemented the results regarding the compression stress-strain curve and the compression Young's modulus of the hydrogels, with the relevant experimental results displayed in Figures 3D and E. The compressive Young's modulus of each hydrogel is comparable, approximately 8.5 kPa.
However, due to the inherent strength limitations of the hydrogel, we were unable to complete the tensile tests. In future research, we will focus on improving the mechanical strength of the hydrogel.
Comments 7: The conclusion should be enriched with some quantitative results.
Response 7: Thank you for pointing this out. We agree with this comment. Therefore, we have further enriched the content of the conclusion (lines 716-729), as follows:
A novel GelMA-based drug-loaded hydrogel with immunomodulatory, antibacterial, and pro-angiogenic properties has been developed as an innovative dressing material to facilitate the rapid healing of skin wounds. This hydrogel dressing is made up of photocrosslinked GelMA and quercetin-loaded ZIF-8. Upon injection into the wound site, GelMA undergoes photocrosslinking to form a gel, demonstrating excellent injectability. Experimental results confirm that the expression levels of M2 polarization-related genes and proteins in the Qu@ZIF-8-Gel hydrogel group were significantly higher than those in the control group, indicating its capability to effectively stimulate M2 macrophage polarization. Additionally, the Qu@ZIF-8-Gel hydrogel significantly inhibited the proliferation of S. aureus and E. coli, achieving an antibacterial efficiency of 99.4 ± 0.3% against E. coli and 98.8 ± 0.6% against S. aureus. Furthermore, the expression levels of angiogenesis-related genes and proteins in the Qu@ZIF-8-Gel hydrogel group were notably increased compared to the control group, and in tube formation assays, the vascular length nearly doubled compared to the control group, suggesting significant angiogenic effects. Subsequent animal studies further validated the immunomodulatory and angiogenic capabilities of Qu@ZIF-8-Gel hydrogel, significantly promoting the repair process of skin wounds. Notably, this hydrogel also exhibits good biocompatibility and ease of synthesis. In summary, the Qu@ZIF-8-Gel hydrogel shows promising potential as a novel dressing material suitable for skin wound repair.
Reviewer 2 Report
Comments and Suggestions for Authors
1. Please provide the source of cells used for in vitro studies.
2. Please describe the cell culture conditions. From which passage did the cells used for the experiment come from. How were the cells obtained for the experiment.
3. Please expand the discussion. The authors did not comment on the obtained results (including IL-10, IL-12, iNOS, the presented wound healing). Please discuss the immunomodulatory and proangiogenic effect of the tested hydrogels in the light of the literature.
Author Response
Comments 1: Please provide the source of cells used for in vitro studies.
Response 1: Thank you for pointing this out. We agree with this comment. Therefore, we have added information regarding the source of the cells. The NIH-3T3, RAW264.7, and HUVEC cell lines were all sourced from the Sunn Biotechnology Co., Ltd (China).
Comments 2: Please describe the cell culture conditions. From which passage did the cells used for the experiment come from. How were the cells obtained for the experiment.
Response 2: Thank you for pointing this out. We agree with this comment. Therefore, we have added a description of cell culture in the methods section (lines 153-160) as per your suggestion, as follows:
Cell culture: Mouse embryonic fibroblast cells (NIH-3T3) were cultured in DMEM media containing 10% calf serum and the recommended dosage of antibiotics (100 U/L penicillin and 100 mg/L streptomycin). Mouse mononuclear macrophages cells (RAW264.7) were cultured in DMEM media supplemented with 10% fetal bovine serum and the recommended dosage of antibiotics. And Human Umbilical Vein Endothelial Cells (HUVEC) were cultured in DMEM/F12 media with 10% fetal bovine serum and the recommended dosage of antibiotics. All cell lines were passaged fewer than five times and incubated at 37°C in a humidified atmosphere with 5% CO2.
Comments 3: Please expand the discussion. The authors did not comment on the obtained results (including IL-10, IL-12, iNOS, the presented wound healing). Please discuss the immunomodulatory and proangiogenic effect of the tested hydrogels in the light of the literature.
Response 3: Thank you for pointing this out. We agree with this comment. Therefore, we have further enriched the discussion section by providing a detailed analysis of the indicators you mentioned above. In particular, we have added supplementary discussions on the experimental results related to the immunomodulation, angiogenesis, and antibacterial properties of the hydrogel. The specific content is as follows:
(lines 623-648) In a previous study, Zhou et al. developed a double-network hydrogel biomaterial composed of snail glycosaminoglycan and GelMA for the treatment of diabetic wounds. Experimental results demonstrated that this biomaterial significantly alleviated inflammation while promoting M2 polarization of macrophages, effectively transitioning diabetic wounds from the inflammatory stage to the proliferative stage, ultimately achieving wound healing [40]. In this study, we observed that ZIF-8-Gel, Qu-Gel, and Qu@ZIF-8-Gel hydrogels upregulated the expression of IL-10 and CD206, which are closely associated with M2 macrophage polarization, while downregulating the expression of IL-12 and iNOS, which are related to M1 macrophage polarization. Furthermore, compared to ZIF-8-Gel and Qu-Gel hydrogels, Qu@ZIF-8-Gel hydrogel exhibited a more pronounced effect on the expression of the aforementioned genes or proteins, with IL-10 levels increasing to 116.7±18.9 pg/mL and the relative expression of the CD206 gene rising to 3.2±0.9, significantly higher than other groups. Conversely, the expression of IL-12 decreased to 28.4±3.0 pg/mL, and the relative expression of the iNOS gene also decreased to 0.5±0.1, markedly lower than the other groups. Subsequent immunofluorescence detection and immunohistochemical staining of tissue samples also revealed the same trend. These results suggest that the zinc ions and quercetin components in Qu@ZIF-8-Gel hydrogel can synergistically promote macrophage polarization toward the M2 phenotype while inhibiting the M1 phenotype, thereby exhibiting more significant immunomodulatory effects. In light of previous research findings, we believe that this regulatory effect on the local inflammatory environment will be beneficial for the rapid healing of skin wounds. Furthermore, subsequent in vivo experiments confirmed our hypothesis, showing that the skin wound healing rate in rats implanted with Qu@ZIF-8-Gel hydrogel significantly accelerated. By day 14 post-implantation, the skin wounds in rats had nearly completely healed, and H&E staining results indicated that the newly formed epidermal tissue had fully covered the wound site, confirming the potential application of Qu@ZIF-8-Gel hydrogel as a wound dressing.
(lines 653-679) Previous research evidence indicates that angiogenesis is crucial for the healing of skin wounds. For example, Guan et al. proposed a sustained oxygenation system composed of oxygen-release microspheres and reactive oxygen species (ROS)-scavenging hydrogel for skin wound treatment. Experimental results confirmed that sustained oxygenation enhanced the expression of growth factors such as vascular endothelial growth factor-A (VEGFA) and platelet-derived growth factor-B (PDGFB) in skin cells, thereby promoting angiogenesis in wounds and ultimately improving wound healing rates [42]. In this study, an important issue to note is that when the concentration of quercetin exceeds 50 μM, it significantly inhibits the expression of vascular endothelial growth factor receptor 2 (VEGFR2) in human umbilical vein endothelial cells (HUVECs) and suppresses tubular structure formation [43], thereby exhibiting anti-angiogenic properties. To address this issue, the present study encapsulated quercetin within ZIF-8, achieving a cumulative re-lease of 16.6±2.1% by day 21 in an environment with a pH of 7.4, thus successfully facilitating its sustained release. Additionally, due to the pro-angiogenic effects of zinc ions, the ZIF-8-Gel hydrogel significantly upregulated the expression of VEGF and HIF-1α genes in HUVECs, substantially promoting the formation of vascular networks. Notably, compared to the control group, the Qu@ZIF-8-Gel hydrogel also significantly upregulated the expression of VEGF and HIF-1α, with relative expression levels of VEGF at 3.6±0.5 and HIF-1α at 3.3±0.8. The capillary length in tube formation assays was also significantly increased, with the Qu@ZIF-8-Gel group showing a length of 21.4±2.8 mm, much higher than the 12.8±3.0 mm observed in the Ctrl group. Subsequent VEGF immunofluorescence results further validated these findings, further confirming that the system effectively mitigated the anti-angiogenic effects of quercetin while harnessing the pro-angiogenic properties of ZIF-8. In subsequent in vivo experiments, the expression of angiogenic markers CD31 and α-SMA in rat skin treated with Qu@ZIF-8-Gel and ZIF-8-Gel hydrogels was significantly increased, further elucidating the pro-angiogenic effect of this hydrogel.
(lines 688-703) Due to the good antibacterial performance of ZIF-8, it is often used as a novel antibacterial agent in the research of infectious disease treatments. For example, Liu et al. encapsulated ZIF-8 in GelMA hydrogel to construct an antibacterial hydrogel system for the treatment of periodontitis. The results showed that this hydrogel significantly reduced bacterial load and exhibited great potential in treating periodontitis [47]. In this study, we encapsulated quercetin within ZIF-8 to achieve enhanced antibacterial effects. The colony-forming unit (CFU) count results showed that compared to the control group and other hydrogel groups, the Qu@ZIF-8-Gel hydrogel effectively inhibited the proliferation of S. aureus and E. coli, achieving antibacterial efficiencies of 99.4 ± 0.3% against E. coli and 98.8 ± 0.6% against S. aureus. This confirms that the Qu@ZIF-8-Gel hydrogel successfully integrates the antibacterial effects of quercetin and ZIF-8, demonstrating superior anti-bacterial performance. Subsequently, SEM results confirmed that the Qu@ZIF-8-Gel hydrogel caused damage to the bacterial membrane structure, further validating previous research findings that both quercetin and ZIF-8 can exert their antibacterial effects by disrupting the integrity of bacterial membrane structures.
Reviewer 3 Report
Comments and Suggestions for Authors
this manuscript on quercetin loaded ZIF nanoparticles for the application of wound healing reveals a detailed and thorough approach to characterizing this system and its potential as a drug delivery device. the scientific methodology is solid, the data presentation is of the highest quality, and the conclusions are supported by the data. the authors have shown a thoughtful approach to the research questions and approached each experiment with the appropriate depth. the only overarching suggestion is to consider some revisions to the busy figures for clarity. the authors have done a fantastic job organizing all the results into very nice looking figures but in some of them (2, 3, and 4 most so), the reader needs to zoom in to maybe 200 or 300% to read the plots. at this zoom level, the draft version images start to lose some integrity and become blurry.
a few more specific suggestions to improve the overall quality are below:
-in figure 2i, the authors should consider revising the axis title to simply degradation %. rate is a confusing term since it implies that it was the speed of degradation that was measured at each time point, and not the actual amount of degradation.
-in figure 2j and 2k, the authors should consider adding a right axis for the actual mass of quercetin and Zn released. the % of loaded is a helpful metric for characterizing the ZIF system in regards to its degradation, release profile, composition, but the biological relevance of the released drug and Zn ions would be aided by real numbers for the masses.
-the authors should consider reformatting figure 4c and 4d. as shown, the confocal scale bars are too small to read and at least in 4c, the field of view is broad enough to make discerning any detail from the images difficult. a tighter crop wouldn't take away from the representative aspect of the image but might help readers see some meaningful difference between what appear to mostly just be black squares on the page.
-the authors should refer to some of the important figures and specific numerical results and conclusions in the discussion. this would help frame the most important points in the study in relation to their real-world importance.
-the authors should also offer a slightly more detailed look at specific existing works in the field during the introduction to contrast to the present work. the novelty of this work is not as clear without contrasting it to the established literature and open questions related to the application of NP drug delivery devices used intended to aid wound healing.
Author Response
Comments 1: in figure 2i, the authors should consider revising the axis title to simply degradation %. rate is a confusing term since it implies that it was the speed of degradation that was measured at each time point, and not the actual amount of degradation.
Response 1: Thank you for pointing this out. We agree with this comment. Therefore, we have revised the axis title to simply degradation (%).
Comments 2: in figure 2j and 2k, the authors should consider adding a right axis for the actual mass of quercetin and Zn released. the % of loaded is a helpful metric for characterizing the ZIF system in regards to its degradation, release profile, composition, but the biological relevance of the released drug and Zn ions would be aided by real numbers for the masses.
Response 2: Thank you for pointing this out. We agree with this comment. Therefore, we have added a right axis for the actual mass of quercetin and Zn released. Additionally, we have made overall modifications to the images to enhance their clarity.
Comments 3: the authors should consider reformatting figure 4c and 4d. as shown, the confocal scale bars are too small to read and at least in 4c, the field of view is broad enough to make discerning any detail from the images difficult. a tighter crop wouldn't take away from the representative aspect of the image but might help readers see some meaningful difference between what appear to mostly just be black squares on the page.
Response 3: Thank you for pointing this out. We agree with this comment. Therefore, we have revised Figures 4C and D to better reflect additional details.
Comments 4: the authors should refer to some of the important figures and specific numerical results and conclusions in the discussion. this would help frame the most important points in the study in relation to their real-world importance.
Response 4: Thank you for pointing this out. We agree with this comment. Therefore, we have revised the discussion section according to your suggestions and added references to key information. For example, in the discussion of the antibacterial results, we have included specific details regarding the antibacterial efficacy. The specific content is as follows:
(lines 693-703) In this study, we encapsulated quercetin within ZIF-8 to achieve enhanced antibacterial effects. The colony-forming unit (CFU) count results showed that compared to the control group and other hydrogel groups, the Qu@ZIF-8-Gel hydrogel effectively inhibited the proliferation of S. aureus and E. coli, achieving antibacterial efficiencies of 99.4 ± 0.3% against E. coli and 98.8 ± 0.6% against S. aureus (Figure 6B, C). This confirms that the Qu@ZIF-8-Gel hydrogel successfully integrates the antibacterial effects of quercetin and ZIF-8, demonstrating superior antibacterial performance. Subsequently, SEM results confirmed that the Qu@ZIF-8-Gel hydrogel caused damage to the bacterial membrane structure (Figure 6D), further validating previous research findings that both quercetin and ZIF-8 can exert their anti-bacterial effects by disrupting the integrity of bacterial membrane structures.
Comments 5: the authors should also offer a slightly more detailed look at specific existing works in the field during the introduction to contrast to the present work. the novelty of this work is not as clear without contrasting it to the established literature and open questions related to the application of NP drug delivery devices used intended to aid wound healing.
Response 5: Thank you for pointing this out. We agree with this comment. Therefore, we have added a more detailed look at the existing research in the introduction section and highlighted the limitations of the existing research to emphasize the innovation of this work. The specific modifications are as follows:
(lines 63-67) To overcome these limitations, researchers have attempted to load quercetin onto drug carriers such as nanoparticles for its sustained release. For example, Bakr et al. loaded quercetin onto chitosan nanoparticles, which effectively improved the bioavailability of quercetin. Additionally, it alleviated cisplatin-induced renal and testicular toxicity by inhibiting oxidative stress, inflammation, and apoptosis [19]. Inspired by the above research, this study aims to encapsulate quercetin within ZIF-8 nanoparticles to prepare Qu@ZIF-8 nanoparticles, leveraging the unique advantages of ZIF-8 to enhance drug stability and control drug release, with the goal of improving the bioavailability of quercetin and effectively prolonging its action time in vivo.
(lines 92-99) Given these advantages, many studies have utilized GelMA hydrogels to construct wound dressings, achieving satisfactory repair results. For instance, in the research con-ducted by Nascimento et al., GelMA hydrogels containing Punica granatum extract were able to promote the differentiation of myofibroblasts, thereby facilitating wound closure [25]. Moreover, in the study by Yi et al., GelMA hydrogels directly encapsulated with quercetin successfully reduced oxidative stress and promoted skin wound healing [26]. Therefore, in this study, we plan to load Qu@ZIF-8 nanoparticles into GelMA hydrogels to construct a novel skin wound dressing.
(lines 100-104) Although various hydrogel-based wound dressing materials have been developed to promote wound healing. However, from a clinical perspective, there are still relatively few wound dressings that can simultaneously ensure ease of application while possessing the characteristics of immunomodulation, infection prevention, and angiogenesis promotion.